# Ecotype Division and Chemical Diversity of *Cynomorium songaricum* from Different Geographical Regions

**DOI:** 10.3390/molecules27133967

**Published:** 2022-06-21

**Authors:** Xinke Zhang, Xiao Sun, Yujing Miao, Min Zhang, Lixia Tian, Jie Yang, Chang Liu, Linfang Huang

**Affiliations:** 1Key Lab of Chinese Medicine Resources Conservation, State Administration of Traditional Chinese Medicine of China, Institute of Medicinal Plant Development, Chinese Academy of Medical Sciences and Peking Union Medical College, Beijing 100193, China; 15013034517@139.com (X.Z.); sunxiao199628@163.com (X.S.); miaoyj1993@163.com (Y.M.); tlxmiao@163.com (L.T.); cliu6688@yahoo.com (C.L.); 2Institute of Medicinal Plant Development, Chinese Academy of Medical Sciences and Peking Union Medical College, Baotou 014040, China; zhangm3426@163.com; 3Tongren Tobacco Company Songtao Branch, Tongren 554100, China; yangdan0027@gmail.com

**Keywords:** *Cynomorium songaricum*, ecotype division, chemical diversity, metabolomics

## Abstract

*Cynomorium songaricum* is an important endangered plant with significant medicinal and edible values. However, the lack of resources and quality variation have limited the comprehensive developments and sustainable utilization of *C. songaricum*. Here, we evaluated the chemical and genetic traits of *C. songaricum* from the highly suitable habitat regions simulated with species distribution models. The PCA and NJ tree analyses displayed intraspecific variation in *C. songaricum*, which could be divided into two ecotypes: ecotype I and ecotype II. Furthermore, the LC-MS/MS-based metabolomic was used to identify and analyze the metabolites of two ecotypes. The results indicated that a total of 589 compounds were detected, 236 of which were significantly different between the two ecotypes. Specifically, the relative content and the kind of flavonoids were more abundant in ecotype I, which were closely associated with the medicinal activities. In contrast, amino acids and organic acids were more enriched in ecotype II, which may provide better nutritional quality and unique flavor. In summary, our findings demonstrate the ecotype division and chemical diversity of *C. songaricum* in China from different geographical regions and provide a reference for the development of germplasm and directed plant breeding of endangered medicinal plants.

## 1. Introduction

*Cynomorium songaricum*, which belongs to the genus *Cynomorium* from the family Cynooriaceae, is a perennial parasitic plant, and is widely distributed in desert regions of Asia, Africa, and Europe [1]. In China, it is generally used as a traditional Chinese medicine for the liver and kidney, to replenish essence and blood, and relax the bowels [2]. Modern pharmacological research has shown that it contains several bioactive constituents such as phenolic acids and flavonoids [3,4], which have the effect of anti-oxidant, anti-viral, anti-obesity, anti-diabetes, anti-Alzheimer, and alleviates memory impairment [5,6,7]. Among all the components, gallic acid, protocatechuic acid, and catechin are the main active ingredients [8]. Meanwhile, it is also considered a valuable food due to its nutritional properties and good taste, which has been developed into *Cynomorium* wine, *Cynomorium* tea, and *Cynomorium* dessert [9]. Therefore, *C. songaricum* has high research value as a food-medicine dual plant.

At present, *C. songaricum* mainly originates from wild resources because of the specific growing and climatic conditions. With the development of natural medicine and health functional foods, the demands of *C. songaricum* keep growing, leading to the depletion of wild resources. In addition, the chemical compositions of *C. songaricum* from different geographical regions are dissimilar, thus affecting differences in the quality [10]. This could be due to adaptation to the specific environment, resulting in the formation of distinct ecotypes. These issues have limited the comprehensive developments and sustainable utilization of *C. songaricum*. However, most previous studies on *C. songaricum* mainly focused on the pharmacological effects and chemical composition. Little research has been done on the suitable habitat distribution, ecotype delimitation, and quality assessment of *C. songaricum*.

Here, chemical and molecular techniques were conjunctively utilized to distinguish *C. songaricum* from different geographical regions: HPLC and DNA barcoding. Recently, because of its faster separation and higher sensitivity, HPLC has been widely used in the content determination of traditional Chinese medicine [11]. DNA barcoding is an alternative technique of species identification [12]. Among these DNA barcodes, internal transcribed spacers (ITS) has been applied widely to clarify its genetic distinctness due to high variability and rich with DNA divergence information [13]. In addition, metabolomics profiling of *C. songaricum* from different origins is crucial for quality evaluation. Widely targeted metabolomics analysis is a rapid, comprehensive, and reliable method for the detection of plant metabolites, which is supported by LC-MS/MS. This technology integrates the great separation efficiency of LC and the structural elucidation and characterization ability of MS [14], which can obtain information about the extensive range of plant metabolites [15]. Currently, metabolic profiling has been successfully used to distinguish between different plant phenotypes, and identify quality-related components [16,17].

In this study, the occurrence records were used to simulate the habitat suitability distribution using the MaxEnt approach, which aimed to determine the major production area. In order to evaluate the properties of *C. songaricum* from different geographical regions, we have used main chemical composition and ITS sequence to distinguish from each other. To further gain insight into the chemical diversity of *C. songaricum*, UPLC-MS/MS-based metabolomic analysis was used to systematically identify and quantify the metabolites in *C. songaricum* from different geographical regions. The results will help to reveal the material basis of nutritional and medicinal value in *C. songaricum*, and provide essential information for the conservation and development of endangered medicinal plants.

## 2. Materials and Methods

### 2.1. Data Sources

The coordinate data of *C. songaricum* were mainly obtained from the following sources: Global Biodiversity Information Facility (GBIF, https://www.gbif.org/, accessed on 18 September 2021), the National Specimen Information Infrastructure (NSII, http://www.nsii.org.cn, accessed on 18 September 2021), and Chinese Virtual Herbarium (https://www.cvh.ac.cn, accessed on 18 September 2021)), and. Google Earth (http://ditu.google.cn/, accessed on 18 September 2021) was used to verify locations and determine the latitude and longitude. All distribution records were saved in CSV format. After duplicate points were removed, there were 78 occurrences collected and used for analysis.

Then, 19 bioclimatic variables were considered as environmental predictors to perform species distribution models [18]. To avoid multicollinearity and obtain reliable and unbiased output, highly correlated variables for species distribution models were removed [19]. Pearson correlation coefficient values were calculated among all the 19 bioclimatic variables and one variable with absolute value greater than 0.8 in each pair was eliminated. Finally, nine bioclimate variables (bio1, bio2, bio3, bio4, bio6, bio9, bio12, bio14, and bio15) were extracted (Appendix A).

### 2.2. Species Distribution Modeling

The habitat suitability of *C. songaricum* was assessed using MaxEnt (v.3.4.1) based on maximum entropy [20]. We used the default Maxent settings with 75% of the data for training and 25% of the data for the testing data set. The number of replicates was set at 10, and these models were used to obtain the final result. Model performance was estimated using area under the curve (AUC) of receiver operator characteristic (ROC) created by MaxEnt. The jackknife test was applied to assess the significance of these variables. Finally, the distribution map was reclassified into four levels: highly suitable, moderately suitable, generally suitable, and not suitable.

### 2.3. Plant Materials

The samples of *C. songaricum* were collected from five highly suitable regions in May, 2017, which was the harvesting period of *C. songaricum*. The succulents were collected from one plant to create one biological sample. The samples as replicates were randomly collected in replicates of three from each region. Detailed information about all samples was given in Appendix A. All samples were identified by Professor Linfang Huang. After excavation, all samples were immediately removed in liquid nitrogen, and then stored at −80 °C until further processing. In this experiment, XJ represents Xinjiang, QH represents Qinghai, GS represents Gansu, NX represents Ningxia, and NM represents Neimenggu.

### 2.4. Quantitative Determination of Gallic Acid, Catechin and Protocatechuic Acid

The standard compounds of gallic acid, protocatechuic acid, and catechin were purchased from Chengdu Must Bio-Tech. Co., Ltd. (Chengdu, China). The proper amount of *C. songaricum* was crushed and sieved using 80 mesh. One gram of each sample was precisely weighed and ultrasonically extracted with 20 mL of methanol solution through ultrasonic extraction for 40 min. The extraction solution was centrifuged for 5 min at 8000 rpm to collect the supernatant, filtered with a 0.22 µm filter. The samples were stored at 4 °C until HPLC analysis, and the injection volume was set at 20 μL. Samples were analyzed with an Atlantis T3 column 4 × 250 mm (Waters, Milford, MA, USA). A gradient elution was performed with the mobile phase of acetonitrile (A) and 0.02% phosphoric acid solution (B). The linear gradient elution program was optimized as follows: 0–5 min, 10%A; 5–26 min, 10%A–13%A; 26–30 min, 13%A–30%A; 30–35 min, 30%A–40%A.

### 2.5. ITS-Based Molecular Identification

The samples taken from dried fleshy stems (25 mg) were ground in a ball mill MM400 (Retsch, Haan, Germany) at a frequency of 30 r/s for 2 min. The total DNA was extracted using Plant Genomic DNA Kits (Tiangen Biotech Co., Ltd., Beijing, China). Primers were designed with Primer Premier 6 (Premier Biosoft, Palo Alto, CA, USA) as follows: AAGGAAGCAGCAGCACATTGAGT (fwd, 5′→3′) and AACCTGCGGAAGGATCATTGTTG (rev, 5′→3′). PCR amplification consisted of 25 µL-volume reaction containing 2.5 µL of 10× PCR buffer, 2 µL of Mg^2+^ (25 mmol/L), 2 µL of dNTPs (2.5 mmol/L), 1 µL of each primer (2.5 µmol/L), 2 µL of template DNA (about 30 ng), and 10 µL of Taq DNA polymerase, the remaining volume consisted of double-distilled water. The products were visualized using 3% agarose gel electrophoresis and directly sequenced used for the Sanger sequencing method with an ABI 3730 sequencer (Applied Biosystems, Waltham, MA, USA). Sequences were manually edited and assembled via CodonCode Aligner 5.1.4 and aligned by MAFFT [21]. Genetic distance and neighbor-joining (NJ) tree were calculated and built with the MEGA v6.0 [22].

### 2.6. Metabolomic Analysis and MS Data

The extraction and detection of samples were performed at the MetWare Biotechnology Co., Ltd (Wuhan, China). The sample extracts were analyzed through LC-ESI-MS/MS system (UPLC, Shim-pack UFLC Shimadzu CBM30A system; MS, 6500 Q TRAP, Applied Biosystems). The UPLC analytical conditions were as follows: column, Waters Acquity UPLC HSS T3 C18 (1.8 µm, 2.1 mm × 100 mm); mobile phase, acetonitrile (0.04% acetic acid, A) and water (0.04% acetic acid, B); gradient system, 95:5 *v*/*v* at 0 min, 5:95 *v*/*v* at 11.0 min, 5:95 *v*/*v* at 12.0 min, 95:5 *v*/*v* at 12.1 min, 95:5 *v*/*v* at 15.0 min; injection volume: 2 μL; flow rate, 0.40 mL/min; temperature, 40 °C. All metabolites were annotated with self-built MetWare database and quantified using multiple reaction monitoring.

Analysis of metabolite data was conducted and processed using MetaboAnalyst v4.0. The resulting three-dimensional matrix contained sample descriptions, *m*/*z* pairs and retention time were analyzed with multivariate data analysis after programmed processing.

### 2.7. Statistical Analysis

The contents of gallic acid, protocatechuic acid, and catechin were sorted and analyzed in triplicate. The results were expressed as the mean ± standard deviation using the IBM SPSS (version 20.0, Chicago, IL, USA). Genetic distances were estimated using the Kimura two-parameter method, and the phylogenetic tree was constructed by the neighbor-joining method.

All the metabolite data were log2-transformed for statistical analysis to improve normality and normalized. The PCA was utilized to depict the overall differences among samples. The PCA involves an orthogonal transformation that converts a group of potentially correlated variables into a group of linearly uncorrelated variables, performed using the prcomp function from the R stats package. The Euclidean distance was used as the cluster method for heatmap-making, and normalized signal intensities of metabolites (unit variance scaling) were visualized as a color spectrum.

Partial least squares-discriminant analysis (PLS-DA) and orthogonal projection to latent structure discriminate analysis (OPLS-DA) was processed to verify the model, and different metabolites between groups were explored [23]. For metabolites with differential accumulation, we selected metabolites with the fold change ≥ 2 (upregulated) or the fold change ≤ 0.5 (downregulated). Then, the variable importance in the projection (VIP) value of the OPLS-DA model was used to screen these differential metabolites with VIP values > 1, which was validated based on a univariate level with adjusted *p* < 0.05. The metabolites with differential accumulation were annotated based on the KEGG database, and then the annotation result was mapped on the KEGG Pathway, which were subjected to pathway enrichment analysis using the online website Metabolite Sets Enrichment Analysis (http://www.msea.ca, accessed on 25 February 2022).

## 3. Results

### 3.1. Prediction of Suitable Habitats for C. songaricum in China

By following the maximum entropy principle of the Maxent model, the maximum possible AUC training value was 0.988 (Appendix A), which showed that the Maxent model could accurately implement the distribution of habitat suitability for *C. songaricum*. The result suggested that the suitable habitats were mainly located in the northwestern region of China, including Xinjiang, Gansu, Neimenggu, Qinghai, Ningxia, Xizang, and Shaanxi (Figure 1). Regarding the habitat suitability class, *C. songaricum* had approximately 980,250 km^2^ highly suitable habitat areas, mainly scattered in Xinjiang, Gansu, Neimenggu, Qinghai, and Ningxia (Table 1).

### 3.2. Ecotype Division of C. songaricum

*C. songaricum* sampling was conducted at five highly suitable habitat areas based on Maxent model results. To allow for a comparison of *C. songaricum* from different geographical regions, the chemical and genetic characteristics were examined based on HPLC analysis and ITS region.

#### 3.2.1. Chemical Characteristics of *C. songaricum* from Different Geographical Regions

Based on the results of HPLC, the contents of gallic acid, catechin and protocatechuic acid of samples from five regions were different (Figure 2). The content of gallic acid in Xinjiang and Qinghai samples was higher than in the other samples, while the content of protocatechuic acid was lower than in others. Additionally, PCA was applied to the above three ingredients’ contents. The results showed that samples from different regions were clustered together and all samples were separated into two main distinct groups, with samples from Xinjiang and Qinghai clustering together, and Gansu, other samples clustering together (Figure 3a).

#### 3.2.2. Genetic Characteristics of *C. songaricum* from Different Geographical Regions

The ITS sequences were used to analyze the genetic diversity among the samples from five different regions. The pairwise genetic distance between all samples ranged from 0.0012 to 0.0073. The largest genetic distance was observed between Neimenggu and Xinjiang, while the closest distance was observed between Ningxia and Gansu (Appendix A). The phylogenetic tree was constructed based on the NJ method. The results showed that all samples fall into two clades, the samples from Xinjiang and Qinghai were grouped into a clade, and others clustered into another clade (Figure 3b). The genetic diversity suggested intraspecific differentiation among *C. songaricum* from different regions.

#### 3.2.3. Two Ecotypes of *C. songaricum* Based on Chemical and Genetic Characteristics

In summary, the classification result based on chemical and genetic characteristics demonstrated a high degree of agreement. From the analysis, *C. songaricum* from different regions were divided into two ecotypes. The samples from Gansu, Neimenggu, and Ningxia shared similar chemical and genetic homogeneity and were classified as ecotype I. The others from Xinjiang and Qinghai were classified as ecotype II (Figure 3).

### 3.3. Metabolite Signatures for the Two Ecotypes

#### 3.3.1. Metabolites Detection in Ecotype I and Ecotype II

We performed LC-MS/MS-based widely targeted metabolomics profiling of ecotype I and ecotype II to detailly study and better understand metabolites differences between two ecotypes. Altogether 589 metabolites were identified, including lots of primary metabolites that may contribute to the nutritional quality, and secondary metabolites that may be attributed to the medicinal quality (Appendix A). Among these metabolites, 540 metabolites both existed in two ecotypes, 38 metabolites were unique to ecotype I, while 11 metabolites were specific to ecotype II (Appendix A).

#### 3.3.2. Differentially Accumulated Metabolites (DAMs) Analysis in Different Ecotypes

To investigate the DAMs between two ecotypes, the PCA analyses were carried out at first. In the PCA plot, the clear separation among two ecotypes and the triplicate samples in each group were distributed on the same side (Figure 4). The results showed that the first principal component (PC1) accounted for 72.5% of the total variance whereas the second principal component (PC2) explained nearly 14.4% of the total variance. This analysis revealed the metabolites between the two ecotypes were different.

To further identify DAMs between two ecotypes, metabolites with the fold change ≥ 2 (upregulated) or ≤0.5 (downregulated) were selected, Then, these metabolites were examined by the OPLS-DA model, and DAMs were separated from insignificant metabolites based on VIP values > 1. In total, 236 DAMs were obtained. Compared with ecotype II, 147 metabolites were upregulated, while 89 metabolites were downregulated in ecotype I (Figure 5a, Appendix A). Hierarchical cluster analysis for the DMs according was shown in the heatmap (Figure 5b). This analysis revealed low biological variability among samples, which could be divided into two major clusters with a different pattern of DMs accumulation. The 236 DMs were classified into 20 different categories, mainly including flavone and its derivates (14%), organic acids (10%), amino acid and its derivatives (9%), nucleotide and its derivates (9%), lipids glycerophospholipids (8%) and others (Figure 5c). In detail, the relative contents and the categories of flavonoids significantly accumulated in the ecotype I. Especially naringenin, phloretin, and apigenin occurred more. While most of the upregulated metabolites in the ecotype II were organic acids, amino acids and its derivatives. Methylglutaric acid, argininosuccinate, citramalate, and l-citrulline accumulated at the significantly higher concentrations in the ecotype II.

#### 3.3.3. KEGG Enrichment of DAMs in Different Ecotypes

For functional annotation, significantly enriched metabolic pathways were determined by KEGG database. The enrichment analysis was performed on DAMs in ecotype I and ecotype II, respectively. The results revealed that 52 metabolites were assigned to 51 pathways from which upregulated in ecotype I. The top 20 significantly enriched metabolic pathway terms were presented in Figure 6a. Of these enrichment pathways, the “isoflavone biosynthesis” (ko00943), “flavonoid biosynthesis” (ko00941), and “phenylpropanoid biosynthesis” (ko00360) were the significant enrichment pathways. In ecotype II, 16 out of the 89 metabolites upregulated in ecotype II were assigned to 40 pathways, and the “biosynthesis of amino acids” (ko01230) and “arginine biosynthesis” (ko00220) were enriched metabolic pathways (Figure 6b). The differences in metabolic pathways between two ecotypes may have certain effects on quality variation.

## 4. Discussion

*C. songaricum* is a precious and endangered medicinal plant. This is the first study to model suitable habitats for *C. songaricum*. The results exhibited the highly suitable habitat areas distributed in Xinjiang, Gansu, Neimenggu, Qinghai, and Ningxia provinces, which was in line with previous records [24]. These regions could be considered to establish artificial cultivation bases for *C. songaricum*.

Based on chemical and genetic features allowed excellent classification of *C. songaricum* from different geographical regions. These results were in good agreement. We observed the existence of two ecotypes of *C. songaricum* in China. The PCA and NJ tree analyses revealed a clear difference in the two ecotypes, and the samples clustered largely according to habitat types, which belong to the same nearby habitats could be clustered together. The variation might be caused by the differences in geographical location and growth environment [25]. Xinjiang and Qinghai were located in Western China, with abundant rainfall and a warm climate. Besides, salt-alkaline soils were widely distributed in these areas [26]. In contrast, Neimenggu, Qinghai, and Ningxia were located in Central West. The climate is dry and cold with a temperature difference between winter and summer. The topography was dominated by desert [27]. Thus, *C. songaricum* diverged into two ecotypes to adapt to diverse climates and environments.

The primary metabolites are directly involved in the growth, development, and reproduction of organisms [28], while secondary metabolites are not essential, although they perform important ecological functions that favor survival. Furthermore, secondary metabolites are a source of medicinal and nutritional resources of great interest to humans [29,30]. The types and quantities of chemical compositions are important factors influencing the quality of medicinal plants, which fluctuated with geographic regions and growth environments. To understand comprehensive metabolic networks of *C. songaricum* from different origins, LC-MS/MS-based widely targeted metabolomics was investigated to identify and quantify all of the metabolites in two types. A total of 589 compounds were detected, 236 of which were significantly different between the two ecotypes. Among these DAMs, 147 compositions were upregulated in ecotype I, and 89 compounds were upregulated in ecotype II, which might provide evidence of metabolic causes underlying the differences between the two ecotypes.

Specifically, the relative content and the kind of flavonoids were more abundant in ecotype I. According to previous reports, these flavonoids exhibited several healthy efficacies including anti-oxidant, anti-aging, and improving blood circulation properties [31], as well as a variety of biological functions such as resisting adversity and protecting plants against ultraviolet radiation [32]. We, therefore, speculate that ecotype I can cause better pharmacological effects and adaptive capacity to the environment. In contrast, the type and quantity of amino acids are primary indicators influencing nutritional quality as well as determining taste [33]. The levels of organic acids are linked closely to sour taste [34]. Amino acids and organic acids were more enriched in ecotype II, which may provide better nutritional quality and unique flavor to ecotype II.

In addition, KEGG pathways were also analyzed between the two ecotypes. The results revealed significant differences in the metabolic patterns between the two ecotypes. In total, 51 and 40 enriched pathways were obtained in ecotype I and ecotype II, to further explain the underlying functions of the DAMs. The DAMs upregulated in ecotype I were enriched in isoflavone biosynthesis pathway, while the DAMs upregulated in ecotype II were enriched in the biosynthesis of amino acids pathway. The results provide novel clues for analyzing the biosynthesis and metabolic pathways of different metabolites in two ecotypes.

## 5. Conclusions

In summary, this is the first study to establish distribution maps and habitat suitability models for *C. songaricum* in China using MaxEnt modeling. Meantime, we successfully divided *C. songaricum* from the different regions into two ecotypes based on chemical and genetic traits. The results revealed that there might be intraspecific variation in *C. songaricum*, and this variation could be related to geographical origin. Furthermore, metabolite diversity and the differences were systematically identified using a widely targeted metabolomics method based on LC-MS/MS. This work provided complete information on the compositions and abundances of metabolites in *C. songaricum*, and revealed that there were distinct metabolic profiles between the two ecotypes. Ecotype I has a higher medicinal value, which can be used as therapeutic drugs and health products. While ecotype II developed into functional foods because of better nutritional quality and unique flavor. Our findings demonstrate the ecotype division of *C. songaricum*, as well as provide a reference for the development of germplasm and directed plant breeding of endangered medicinal plants.

## Figures and Tables

**Figure 1 molecules-27-03967-f001:**
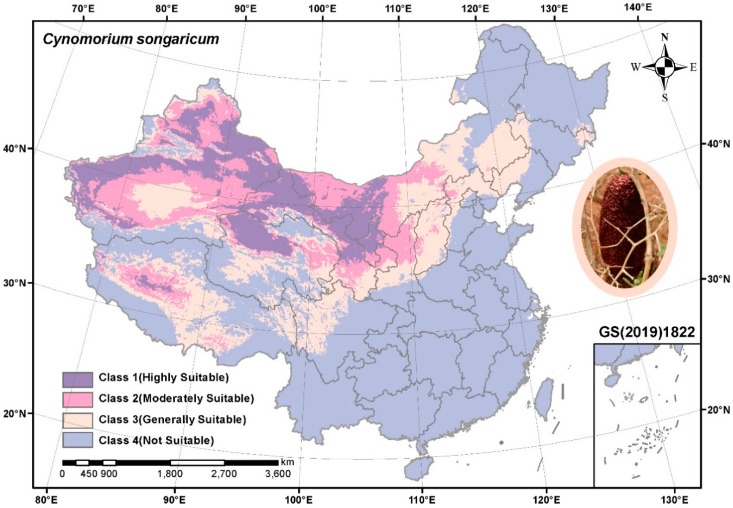
Habitat suitability distribution of *C. songaricum* in China. Map figure number: GS(2019)1822.

**Figure 2 molecules-27-03967-f002:**
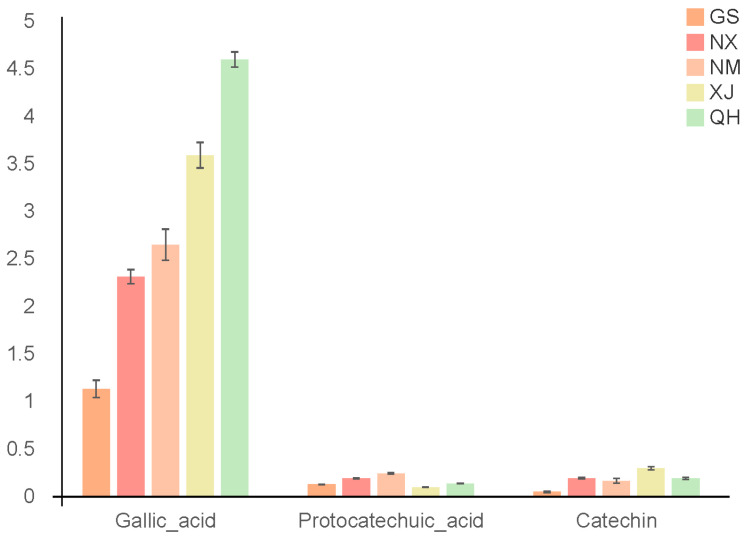
Contents of three chemical components in different geographical populations of *C. songaricum*.

**Figure 3 molecules-27-03967-f003:**
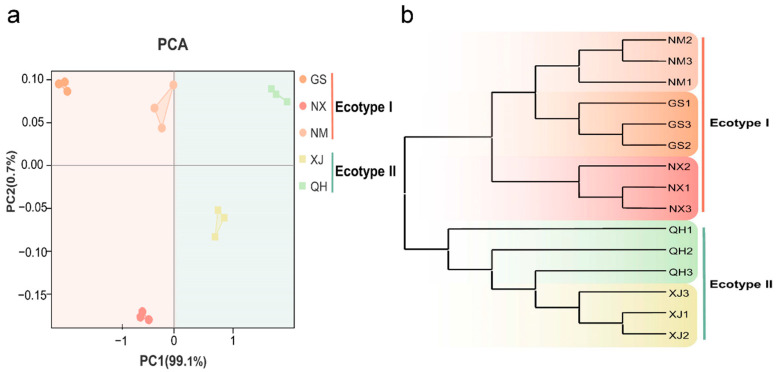
Analysis of characteristics in two ecotypes of *C. songaricum*. (**a**) PCA analysis of chemical characteristics. (**b**) NJ tree based on genetic characteristics.

**Figure 4 molecules-27-03967-f004:**
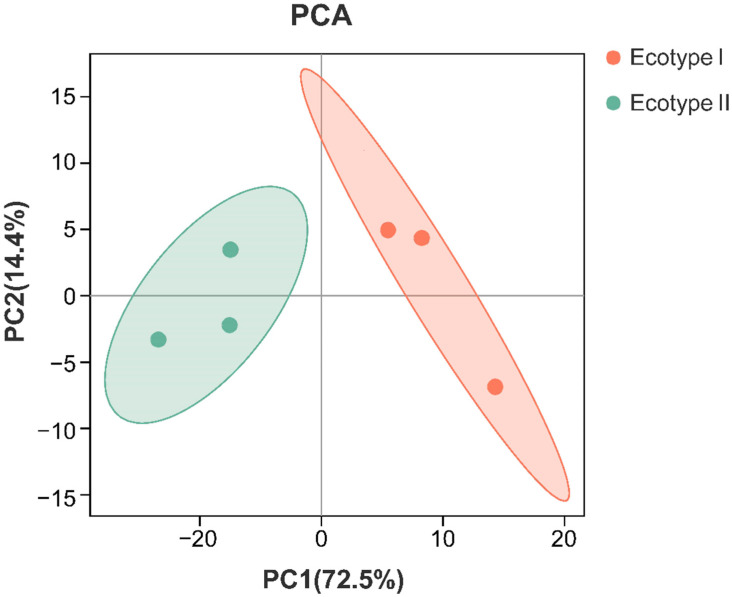
PCA score plot based on all metabolites identified in two ecotypes.

**Figure 5 molecules-27-03967-f005:**
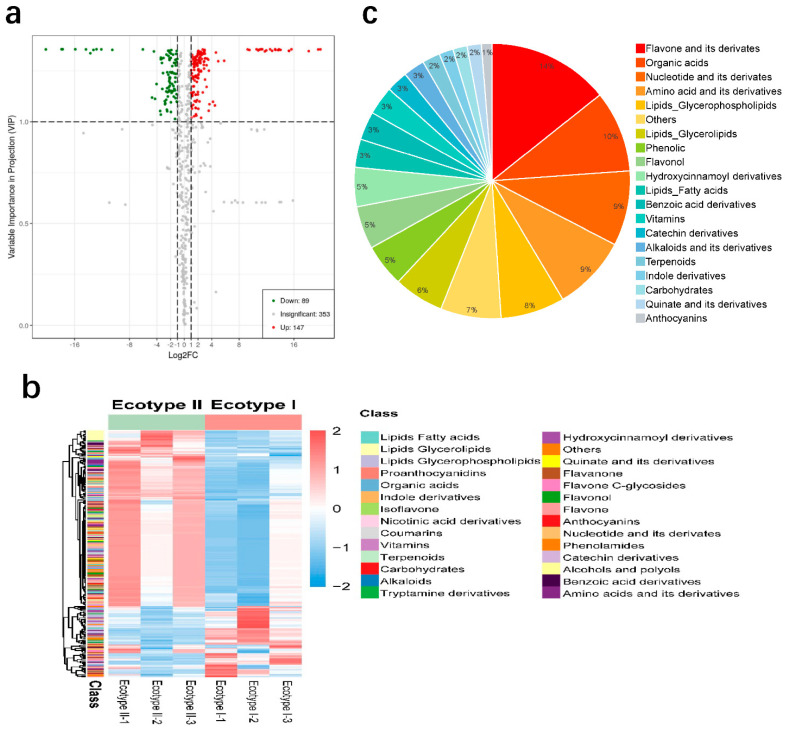
Differentially Accumulated Metabolites (DAMs) between two ecotypes. (**a**) Volcano plot of the 589 metabolites identified. DAMs were defined as metabolites with fold change ≥ 2 (upregulated) or ≤0.5 (downregulated) in ecotype I and ecotype II. A threshold of VIP ≥ 1 was used to separate differential metabolites from insignificant metabolites. (**b**) Heatmap of 236 DMs accumulation pattern. The color indicates the level of accumulation of each metabolite, from low (blue) to high (red). (**c**) Pie chart depicting the biochemical categories of the DAMs identified between eotype I and eotype II.

**Figure 6 molecules-27-03967-f006:**
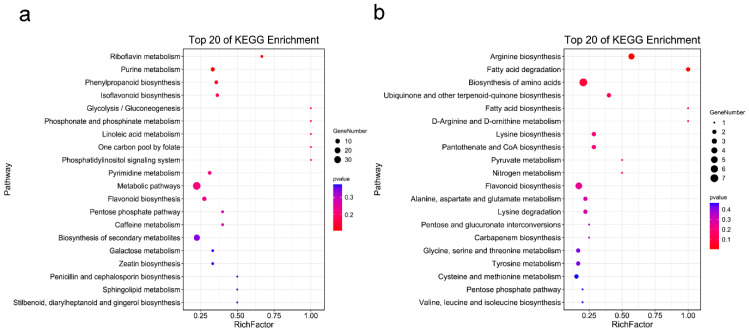
The top 20 of KEGG enrichment of DAMs in (**a**) eotype I and (**b**) eotype II, respectively.

**Table 1 molecules-27-03967-t001:** Different suitable regions of *C. songaricum* in China.

No.	Region	Suitability Scale			
		Highly Suitable Area (km^2^)	Moderately Suitable Class (km^2^)	Generally Suitable Class (km^2^)	Not Suitable Class (km^2^)
1	Xinjiang	489,575	532,125	271,725	230,125
2	Gansu	187,875	105,075	87,500	22,700
3	Neimenggu	155,775	228,200	341,100	338,425
4	Qinghai	115,475	80,450	189,275	282,225
5	Ningxia	31,550	16,800	675	0
6	Xizang	20,500	110,975	385,150	669,500
7	Shaanxi	675	60,100	66,475	70,375

## Data Availability

Not applicable.

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
