# Peer review of "Ecotype Division and Chemical Diversity of *Cynomorium songaricum* from Different Geographical Regions"

_molecules, 2022, doi:10.3390/molecules27133967_

Round 1

Reviewer 1 Report

The presented work is an interesting comprehensive study on the interesting topic of biochemical variability of the valuable medical plant Cynomorium songaricum. This is an important topic in practice, as the pharmaceutical potential of plants often varies by geography and can be lost when cultivation is attempted. The authors compared geographic, genetic and metabolomic similarities. Such an integrated approach gives the work a special value. This work looks as complete finished investigation, suitable for publication. However, there are the several questions and comments to be considered and answered:

  1. The details of statistical analysis should be described in more details. It is necessary to describe data preprocessing: missing data imputation, normalization e. t. c. Distance metrics and clustering method for all dendrograms (including metabolite clustering in the heatmap) should be given.
  2. Authors present interesting data on the content of catechin, gallic and protocatechuic acids, but PCA on three variables is not a good idea. In my opinion, it would be better to present the samples in the space with the concentrations as coordinates (3D or three 2D plots). Alternatively, figure S2 could be placed in the main part of the article, and provided with an indication of the statistical significance of the differences.
  3. Data presentation could be improved. In Figure 3a, authors need to indicate the nature of the replication (origins?). Heatmap 3b is not informative. It might be better to put the heatmap after the DAMs detection and use only them (may be top rated, or grouped) with names. Table of raw metabolite levels need to be given in supplementary.
  4. A small number of typos and stylistic inaccuracies are also present. Line 221 and 222 are broken by figure and one word is left alone. Is the capitalized “F” in the name of professor LinFang Huang correct?

Author Response

Manuscript ID: molecules-1747624

Title: Ecotype division and chemical diversity of Cynomorium songaricum from

different geographical regions

Dear Reviewer,

We would like to express our sincere thanks to you for giving us an opportunity to revise our manuscript entitled “Ecotype division and chemical diversity of Cynomorium songaricum from different geographical regions” (ID: molecules-1747624). We have revised according to the reviewers’ constructive comments, which are marked in green in the paper. Below, please find our point-by-point responses to the reviewers’ comments.

We have read the referee’s comments very carefully and we have revised, updated, and added the content, sentences, words, and references of the article according to the comments.

Again, we would like to express our great appreciation to reviewers for comments on our paper and we have revised our manuscript.

Reviewer 2 Report

Dear authors,

Here are some comments that should be taken into account before publishing this manuscript:

Scientific names of species should be in italics (line 175, 190, 199, Table S2, etc.). Check in the bibliography for the names of the species.

Adapt the letter format to the indications of the journal (line 171,172,188, etc).

References section: Change font format. Scientific names in italics. Journal names must be abbreviated.

Add a paragraph between the point 3.2 and 3.2.1.

The section " Author Contributions" must be included.

Line 32: I don't understand the sentence inside the parentheses. What is the relationship between terpenes and gallic acid? should be clarified.

Line 42, 47, 56...: The authors use the word "quality" throughout the text. This term is very relative. It should be explained better.

Line 53: ITS. They must describe the acronym the first time.

Line 65-73: The introduction does not clearly describe the objectives or hypotheses. The information contained in these lines should not appear in the introduction since it corresponds to material and methods and results. It should be replaced by a description of the objectives of the work.

Line 86: “we examined”: Do not use personal forms.

Line 100-106: The authors must explain the reason for choosing these 5 regions. The authors should describe more details about the phenological stage of the plant, date of collection, etc. Also about the harvesting method, etc.

Line 107: Authors should describe the three main compounds.

Line 177-179: What is described in these three lines should have been explained in the introduction and in material and methods.

Line 277-278: The authors should not explain the differences between primary metabolism and secondary metabolism in this way. I wonder if the primary metabolites aren't also important from a human utilization. It must be expressed in another way.

Figure 1: The figure caption should explain GS(2019)1822 ??

Line 259: I do not know if the species is endangered or has some level of protection. If so, the authors should present the relevant certificates to justify their collection.

Author Response

Manuscript ID: molecules-1747624

Title: Ecotype division and chemical diversity of Cynomorium songaricum from

different geographical regions

Dear editor,

We would like to express our sincere thanks to you for giving us an opportunity to revise our manuscript entitled “Ecotype division and chemical diversity of Cynomorium songaricum from different geographical regions”. (ID: molecules-1747624).

We have made revision according to the editor’s and reviewer’s constructive comments, which marked in green in the paper. Below, please find our point-by-point responses to the editor’s and reviewer’s comments.

  Again, we would like to express our great appreciation to you for comments on our paper.

Round 2

Reviewer 2 Report

Dear authors,

Thank you for including the suggestions made in the new manuscript. On this occasion, I just want to make two suggestions that I hope will be incorporated into the final manuscript.

The first observation refers to point 6 of my previous review. The authors continue to define gallic acid and protocatechuic acid as triterpene compounds. These compounds are simple phenolic acids. Gallic acid (3,4,5-trihydroxybenzoic) is used as a standard to quantify phenols. This statement cannot be accepted.

The second observation is related to the answer made by the authors on the differences between primary and secondary metabolism (lines 295-296). The new sentence is equally incorrect. I propose the following: Primary metabolites are directly involved in the growth, development and reproduction of organisms, while secondary metabolites are not essential, although they perform important ecological functions that favor survival. Furthermore, secondary metabolites are a source of medicinal and nutritional resources of great interest to humans.

Best regards

Author Response

Manuscript ID: molecules-1747624

Title: Ecotype division and chemical diversity of Cynomorium songaricum from

different geographical regions

Dear Reviewer,

We would like to express our sincere thanks to you for giving us an opportunity to revise our manuscript entitled “Ecotype division and chemical diversity of Cynomorium songaricum from different geographical regions” (ID: molecules-1747624). We have revised according to the reviewers’ constructive comments, which are marked in green in the paper. Below, please find our point-by-point responses to the reviewers’ comments.

We have read the referee’s comments very carefully and we have revised, updated, and added the content, sentences, words, and references of the article according to the comments.

Again, we would like to express our great appreciation to reviewers for comments on our paper and we have revised our manuscript.

Major comments:

Point 1. The first observation refers to point 6 of my previous review. The authors continue to define gallic acid and protocatechuic acid as triterpene compounds. These compounds are simple phenolic acids. Gallic acid (3,4,5-trihydroxybenzoic) is used as a standard to quantify phenols. This statement cannot be accepted.

Response: Thank you for pointing this out, the mistake has now been corrected. We have now corrected this error. Please refer to the line 33 in manuscript. The detailed modified content as follows:

Modern pharmacological research has shown that it contains several bioactive constituents such as phenolic acids and flavonoids [3, 4], which has the effect of anti-oxidant, anti-viral, anti-obesity, anti-diabetes, anti-Alzheimer, and alleviates memory impairment.

Point 2. The second observation is related to the answer made by the authors on the differences between primary and secondary metabolism (lines 295-296). The new sentence is equally incorrect. I propose the following: Primary metabolites are directly involved in the growth, development and reproduction of organisms, while secondary metabolites are not essential, although they perform important ecological functions that favor survival. Furthermore, secondary metabolites are a source of medicinal and nutritional resources of great interest to humans.

Response: Thank you for your valuable comments on our manuscript which we have amended according to your suggestions. Please refer to the line 296-300 in manuscript. The detailed modified content as follows:

The primary metabolites are directly involved in the growth, development and reproduction of organisms [28], while secondary metabolites are not essential, although they perform important ecological functions that favor survival. Furthermore, secondary metabolites are a source of medicinal and nutritional resources of great interest to humans [29, 30].
